# Pyroptosis: A Newly Discovered Therapeutic Target for Ischemia-Reperfusion Injury

**DOI:** 10.3390/biom12111625

**Published:** 2022-11-03

**Authors:** Yu Zheng, Xinda Xu, Fanglu Chi, Ning Cong

**Affiliations:** 1Department of Otorhinolaryngology, Eye Ear Nose and Throat Hospital, Fudan University, Shanghai 200031, China; 2Shanghai Clinical Medical Center of Hearing Medicine, Shanghai 200031, China; 3NHC Key Laboratory of Hearing Medicine, Fudan University, Shanghai 200031, China; 4Research Institute of Otorhinolaryngology, Fudan University, Shanghai 200031, China

**Keywords:** pyroptosis, ischemia-reperfusion injury, gasdermin, therapeutic target

## Abstract

Ischemia-reperfusion (I/R) injury, uncommon among patients suffering from myocardial infarction, stroke, or acute kidney injury, can result in cell death and organ dysfunction. Previous studies have shown that different types of cell death, including apoptosis, necrosis, and autophagy, can occur during I/R injury. Pyroptosis, which is characterized by cell membrane pore formation, pro-inflammatory cytokine release, and cell burst, and which differentiates itself from apoptosis and necroptosis, has been found to be closely related to I/R injury. Therefore, targeting the signaling pathways and key regulators of pyroptosis may be favorable for the treatment of I/R injury, which is far from adequate at present. This review summarizes the current status of pyroptosis and its connection to I/R in different organs, as well as potential treatment strategies targeting it to combat I/R injury.

## 1. Introduction

Ischemia-reperfusion (I/R) injury is caused by the resumption of blood supply (reperfusion) to an organ or tissue after a period of restricted blood supply, resulting in pathological processes that can lead to cell death and organ dysfunction. I/R injury can occur in many vital organs, including the brain, heart, and kidneys; it can also occur in less noticed organs, such as the cochlea and vestibule in the ear. The mechanisms of I/R injury comprise oxidative stress, sterile inflammation, calcium overload, mitochondrial dysfunction, and activation of various cell death pathways—including, but not limited to, apoptosis, necrosis, and autophagy [1]. Pyroptosis is another type of regulated cell death, characterized by cell membrane pore formation, pro-inflammatory cytokine release, and cell lysis [2]. It has previously been found that pyroptosis is closely related to the development of tumors [3]. Furthermore, recent studies have shown that pyroptosis also plays an important role in I/R injury. Because the current treatment and management of I/R injury are far from satisfactory [4], advances in understanding of the mechanisms of pyroptosis and its role in I/R injury may lead to innovative therapeutic strategies for treating patients with I/R-associated tissue inflammation and organ dysfunction. Therefore, in this review, we summarize the mechanisms of pyroptosis and its role as a potential therapeutic target for I/R injury.

## 2. Overview of Pyroptosis

Pyroptosis was first discovered by Zychlinsky et al. in 1992. They reported that the death of *Shigella flexneri*-infected macrophages was dependent on caspase-1 rather than caspase-3, which was then thought to be only involved in apoptosis [5]. In 1996, Chen et al. found that the invasion plasmid antigen B of *Shigella flexneri* could bind to the ICE (interleukin-1β-converting enzyme, caspase-1) directly, leading to the activation of enzymes in the infected macrophages and ultimately resulting in the death of those infected macrophages [6]. In 1999, Hersh et al. discovered that caspase-1 plays an indispensable role in this specific cell death mode, as caspase-1-knockout macrophages could not be induced into death by *Shigella flexneri* infection [7]. In 2001, this form of regulated cell death was named “pyroptosis” by Cookson and Brennan after they found a similar phenomenon in macrophages infected with *Salmonella typhimurium* [8]. Seven years later, fragmented DNA and a damaged cell membrane were found by Cookson et al. during the process of pyroptosis, which led to the release of intracellular content and triggered a severe inflammatory response [9]. In 2011, Kayagaki et al. found that caspase-11 was able to induce mouse macrophage death, which is essentially similar to caspase-1-mediated pyroptosis. They termed this caspase-11-mediataed pyroptosis as the “non-canonical pathway of pyroptosis” [10].

In 2015, Shao et al. conducted CRISPR-Cas9 nuclease screening for the whole genome of caspase-11- and caspase-1-mediated pyroptosis of mouse bone marrow macrophages, and they eventually identified gasdermin D (GSDMD) as the inflammatory caspase substrate through which caspases activated pyroptosis [11]. This study further proved that caspase-1 and caspase-4/5/11 specifically cleaved the linker between the amino-terminal gasdermin-N and the carboxy-terminal gasdermin-C domains in GSDMD, ultimately leading to cell pyroptosis, which was triggered only by the gasdermin-N domain.

GSDMD is a member of the human gasdermin superfamily, which consists of gasdermin A/B/C/D (GSDMA/B/C/D), gasdermin E (GSDME, also known as DFNA5), and DFNB59 (PJVK); the mouse gasdermin superfamily is composed of GSDMA1-3, GSDMC1-4, GSDMD, GSDME, and PJVK [12,13,14,15]. Among these proteins, GSDMD and GSDME are currently the most comprehensively researched in terms of pyroptotic death. Moreover, other than PJVK, these conserved proteins consist of two conserved domains, the N-terminal pore-forming domain (PFD) and the C-terminal repressor domain (RD) [16,17,18]. Most of the gasdermins’ N-terminals can trigger pyroptotic death, but this has not been found in PJVK so far [19,20]. In general, gasdermins maintain oligomerization through the interaction of two domains. When cells are stimulated by various factors, gasdermins are cleaved by certain caspases into the N-terminal PFD and the C-terminal RD. After dissociation, the PFD oligomerizes on the cell membrane to form pores and cause cell pyroptosis [21,22].

## 3. Mechanisms of Pyroptosis

It was formerly believed that there are two pathways of pyroptosis: one is the canonical pathway dependent on caspase-1, and the other is the non-canonical pathway dependent on human caspase-4/5 or mouse caspase-11. However, due to in-depth study in recent years, other pathways of pyroptosis have been found (Figure 1). The detailed mechanisms of pyroptosis and its pathways are discussed below.

### 3.1. Canonical Pathway

The canonical pathway of pyroptosis begins with cytoplasmic pattern recognition receptors (PRRs, also known as inflammasome receptors) recognizing pathogen-associated molecular patterns (PAMPs, e.g., glycans, lipopolysaccharides (LPSs)) or danger-associated molecular patterns (DAMPs, e.g., fibrinogen, heat shock proteins, reactive oxygen species (ROS)) [23,24]. With the stimuli of signal molecules such as bacteria or ROS, PRRs assemble with pro-caspase-1 and ASC to form inflammasomes [25,26,27]. Although a variety of PRRs, like NOD-like receptors (NLRs) and Toll-like receptors (TLRs), are involved in this process, only a subset of PRRs—including NLR family pyrin domain-containing (NLRP)1, NLRP3, and NLRP4; absent in melanoma 2 (AIM2); and pyrin—is known to be able to directly assemble inflammasomes [28]. Most inflammasomes are composed of the following: (i) leucine-rich repeat-containing proteins (NOD-like receptors, NLRs); (ii) pro-caspase-1; and (iii) the adapter apoptosis-associated speck-like protein containing a caspase recruitment domain (CARD) (ASC) [29,30,31,32]. More specifically, ASC contains a pyrin domain (PYD) and a caspase activation and recruitment domain (CARD) [28]. CARD is indispensable in recruiting pro-caspase-1 to form the inflammasome [28]. The precursor of caspase-1 (pro-caspase-1) is activated after the assembly of the inflammasome and is hydrolyzed into two fragments to form a dimer and become mature caspase-1 [33]. On the one hand, the mature caspase-1 can cleave the executive protein GSDMD at Asp275 to form the 22kDa C-terminal repressor domain (RD) and the 32kDa pore-forming domain (PFD). Then, the N-terminal PFD oligomerizes and penetrates the cell membrane, forming non-selective pores with pore sizes of about 18 nm [34]; the ensuing increase in osmotic pressure then causes an influx of water followed by cell swelling and the release of cytokines and various cytoplasmic contents. On the other hand, the mature caspase-1 also lyses the precursors of IL-1β and IL-18 (pro-ILβ and pro-IL-18, respectively) to form mature IL-1β and IL-18, which are then released through holes formed by the N-terminal PFD and lead to cell pyroptosis [35]. However, it is worth noting that a recent study examined the cryo-electron microscopy structures of the pore and the prepore of GSDMD. The research revealed the different conformations of the two states and indicated that gasdermin D-mediated pores may not be non-selective and may favor the passage of IL-1β and IL-18 over that of their precursors [36]. Of course, further studies are needed to verify the selectivity or non-selectivity of gasdermin D-mediated pores.

### 3.2. Non-Canonical Pathway

Unlike the canonical pathway, human caspase-4/5 and mouse caspase-11 can directly bind to bacterial lipopolysaccharides (LPSs), thus bypassing the need for inflammasome sensors and resulting in cell pyroptosis [37]. Similar to mature caspase-1, activated caspase-4/5 or caspase-11 can also cleave GSDMD, thus releasing the N-terminal PFD. Then, the oligomerized N-terminal PFD is transferred to the cell membrane, leading to the formation of cell membrane pores [38]. The disruption of osmotic balance results in the release of cell contents and cell swelling and bursting, ultimately leading to cell pyroptosis. In addition, activated caspase-11 can not only promote K+ efflux through the induction of GSDMD cell membrane pore formation, but it can also activate the pannexin-1/ATP/P2 × 7 pathway by cleaving the pannexin-1 channel, releasing ATP, and activating the P2X7 receptor to cause ATP-induced loss of intracellular K+ and NLRP3 inflammasome activation. Once the NLRP3 inflammasome is assembled, the canonical pyroptosis pathways can be activated. Although caspase-4/5/11 cannot cleave pro-IL-1β or pro-IL-18 by themself, they can activate the pannexin-1/ATP/P2X7 pathway, thus leading to the activation of the canonical NLRP3/caspase-1 pathway and the ensuing maturation and secretion of IL-1β/IL-18 [39,40,41,42].

### 3.3. Caspase-3/8-Mediated Pathway

Caspase-3 has previously been considered as a marker and key molecule of apoptosis, but many studies have shown that caspase-3 is also involved in the process of pyroptosis. Under the stimulation of tumor necrosis factor-α (TNF-α) or chemotherapy drugs, caspase-3 has been shown to specifically cleave gasdermin E (GSDME), thus releasing the N-terminal PFD of GSDME [43]. The oligomerized N-terminal PFD of GSDME is then transferred to the cell membrane, forming non-selective pores and inducing cell pyroptosis. In addition, it is intriguing to note that activated caspase-3 would induce cell pyroptosis when the GSDME expression level was high, while activated caspase-3 would induce cell apoptosis when the GSDME expression level was low [19]. Moreover, it has been found that pathogenic *Yersinia* can inhibit TGF (transforming growth factor)-β-activated kinase 1 (TAK1) in mouse macrophages via the effector protein YopJ. As a result, receptor-interacting protein kinase 1 (RIPK1) and caspase-8 are activated. Subsequently, caspase-8 cleaves GSDMD and GSDME, forming the “gasdermin channel” on the plasma membrane that mediates pyroptosis and exacerbates the inflammatory response [44,45].

### 3.4. Granzyme-Mediated Pathway

A 2020 study found that CAR T cells could activate caspase-3 in target cells by releasing granzyme B (GzmB), thus activating the caspase-3/GSDME-mediated pyroptotic pathway [46]. Later, it was proven that GzmB can directly cleave GSDME and induce pyroptosis [47]. Meanwhile, it was also reported that natural killer cells and cytotoxic T lymphocytes killed GSDMB-positive cells by pyroptosis. The killing effect stemmed from GSDMB cleavage at the Lys229/Lys244 site by lymphocyte-derived granzyme A (GzmA) [48]. This was the first study to show that gasdermin could be hydrolyzed at non-aspartic acid sites and form pores, overturning the former understanding that pyroptosis can only be activated by caspases.

## 4. Pyroptosis and I/R Injury

I/R injury mainly revolves around various kinds of cell death and subsequent organ dysfunction. Previous studies have identified that necrosis, apoptosis, and autophagy-associated cell death are implicated in I/R injury [1]. Consequently, treatments targeted at these cell death patterns have also been investigated. Though some progress has been made in dealing with I/R injury so far, the outcomes are still far from satisfactory. Fortunately, the discovery of pyroptosis has brought new hope and impetus to the treatment of I/R injury, as an increasing number of studies have suggested that pyroptosis is involved in I/R injury (the link between pyroptosis and I/R injury was shown in Figure 2). This could allow the adoption of potentially feasible and effective approaches to alleviating I/R injury by targeting pyroptotic cell death. This section discusses the role of pyroptosis in I/R injury in different organs and possible therapeutic strategies to attenuate the intensity of I/R injury.

### 4.1. Pyroptosis and Cerebral I/R Injury

Cerebral artery occlusion is one of the main causes of ischemic strokes, and reperfusion is the best method to restore blood supply to cerebral ischemia and protect the brain from ischemic injury. However, reperfusion may cause more severe secondary injury to brain tissue, which is known as cerebral I/R injury. In 2019, a study conducted by Zhang et al. demonstrated that pyroptosis was involved in cerebral I/R injury, with gasdermin D acting as the key executor of pyroptosis in their model [49]. In 2020, Sun Rui et al. established an ischemic stroke mouse model via middle cerebral artery occlusion [50]. At the same time, the primary cortical neurons were extracted and cultured, and oxygen–glucose deprivation was used to establish an in vitro model. Their study showed that a low-density lipoprotein receptor (LDLR) inhibits neuronal death after cerebral I/R injury by inhibiting NLRP3-mediated neuronal pyroptosis. Similarly, other studies have also shown that hispidulin, glycosides, remimazolam, and dendrobium alkaloids can all exert neuroprotective effects in the event of cerebral I/R injury by inhibiting NLRP3-mediated pyroptosis [51,52,53,54]. In addition, it has been reported that valproic acid and overexpression of CHRFAM7A as well as down-regulation of X-box binding protein l (XBP-1) can also alleviate cerebral I/R injury by inhibiting the caspase-1-mediated canonical pathway of pyroptosis [55,56,57].

Meanwhile, microRNAs are a class of small non-coding RNAs that play a significant role in various biological processes related to cell death. A recent study has suggested that microRNA-124 inhibits pyroptosis induced by cerebral I/R injury by regulating the STAT3 signaling pathway [58]. Furthermore, it has been suggested that ginsenoside Rd could attenuate cerebral I/R injury by exerting an anti-pyroptotic effect via the miR-139-5p/Keap1/Nrf2 axis [59]. Moreover, exosomes are small vesicles containing complex RNAs and proteins involved in cell-to-cell communication; a study indicated that exosomes derived from bone marrow mesenchymal stem cells attenuate cerebral I/R injury-induced neuroinflammation and pyroptosis by modulating microglia M1/M2 phenotypes [60]. Additionally, salvianolic acid injection also alleviates cerebral I/R injury by regulating the M1/M2 phenotypes of microglia and inhibiting the NLRP3 inflammasome/pyroptosis axis in microglia [61]. Finally, mircornA-29A in astrocyte-derived extracellular vesicles was found to inhibit cerebral I/R injury through TP53INP1 and the NF-κB/NLRP3 axis [62]. Here, it is worth noting that berberine also has a protective effect against neuron pyroptosis induced by cerebral I/R via inhibiting the NF-κB/NLRP3 pathway [63]. In addition to these drugs that can attenuate cerebral I/R injury, some researchers have even found that exercise and electroacupuncture can exert neuroprotective effects against cerebral I/R injury by alleviating inflammasome-induced pyroptosis [64,65]. While all the aforementioned studies focused on how to suppress cerebral I/R injury, some studies have focused on the opposite. For instance, Liang et al. found that the long non-coding RNA MEG3 can promote cerebral I/R injury by increasing pyroptosis via the targeting of the miR-485/AIM2 axis [66]. In a nutshell, these findings support the hypothesis that pyroptosis is involved in brain I/R injury and can be a new therapeutic target for treating cerebral I/R injury.

### 4.2. Pyroptosis and Myocardial I/R Injury

I/R injury is the most common pathological type of cardiovascular disease, which is often seen in heart transplantation, thrombolytic therapy, cardiopulmonary bypass surgery, and coronary artery bypass grafting. As early as 2017, it was found that cytoprotective activated protein C could avert NLRP3 inflammasome-induced myocardial I/R injury by inhibiting mammalian target of rapamycin complex 1 (mTORC1) [67]. In the same year, another study illuminated that pyroptosis mediated by the activation of the NLRP3 inflammomere aggravated myocardial I/R injury in diabetic rats [68]. In addition, it was also reported that calpain silencing inhibits the activation of the canonical pyroptotic pathway (NLRP3/ASC/caspase-1) and reduces myocardial I/R injury in mice [69]. Furthermore, Ye et al. also illustrated that emodin could alleviate gasdermin D-mediated myocardial I/R injury by inhibiting the TLR4/MyD88/NF-κB/NLRP3 inflammasome pathway [70]. Moreover, Ding et al. reported that by inhibiting microRNA-29A, SIRT can be targeted to inhibit oxidative stress and NLRP3-mediated pyroptosis to alleviate myocardial I/R injury [71]. Similarly, by down-regulating microRNA-29B, dexmedetomidine is able to inhibit cell pyroptosis caused by myocardial I/R injury in rats [72]. Furthermore, Dai et al. demonstrated that M2 macrophage-derived exosomes carry microRNA-148A to inhibit the TXNIP and TLR4/NF-κB/NLRP3 signaling pathways, thereby alleviating myocardial I/R injury [73]. Liu et al. illustrated that cardiac fibroblast-derived exosomes carry microRNA-133A to suppress cardiomyocyte pyroptosis in myocardial I/R injury, and Zhang et al. indicated that hypoxic cardiac microvascular endothelial cell-derived exosomes carry microRNA-27b-3p to alleviate myocardial I/R Injury via the Foxo1/GSDMD signaling pathway [74,75]. Similar to their functions in cerebral I/R injury, the exosomes of mesenchymal stem cells can also mitigate myocardial I/R injury through inhibiting the canonical pathway of pyroptosis [76,77]. In addition, by targeting the Akt/GSK3β/NF-κB pathway, aesculin can inhibit NLRP3 inflammasome-mediated pyroptosis and thereby attenuate myocardial I/R injury [78]. Apart from the aforementioned drugs, ethyl acetate extract of cinnamomi ramulus, cinnamic acid, igramomod, piperazine ferulate, sevoflurane, trimetazidine, tubastatin, geniposide, intracellular ion channel inositol 1,4,5-triphosphate receptor (IP3R1), and a soluble receptor for late glycation end products have all been found to have protective effects against myocardial I/R injury [79,80,81,82,83,84,85,86,87,88]. Additionally, sweroside can also protect against myocardial I/R injury partly by regulating the Keap1/Nrf2/NLRP3 axis to inhibit oxidative stress and pyroptosis [89]. However, both aquaporin 4 and uric acid can aggravate myocardial I/R injury via NLRP3 inflammasome-mediated pyroptosis [90,91]. Moreover, the circRNA circ-NNT promoted myocardial I/R injury through activating pyroptosis by sponging miR-33a-5p and regulating ubiquitin-specific protease-46 (USP46) expression [92]. In addition, it has been proven that methyltransferase-like protein 3 (METTL3) can aggravate myocardial I/R injury via facilitating DGCR8 binding to pri-miR-143-3p through m6A modification, thus enhancing miR-143-3p expression to suppress protein kinase C epsilon transcription and further exacerbate cardiomyocyte pyroptosis [93]. Therefore, targeting myocardial pyroptosis may provide a new therapeutic strategy for patients with cardiovascular recanalization as well as heart transplant recipients.

### 4.3. Pyroptosis and Renal I/R Injury

The earliest study on pyroptosis and renal I/R injury, which can be traced back to 2014, found that renal I/R injury can trigger the pyroptosis of renal tubule epithelial cells through the CHOP-caspase-11 pathway [94]. Although it had not yet been confirmed that the gasdermin family was the executor of pyroptosis at that time, this study found that levels of pyroptosis-related proteins, including caspase-1, caspase-11, and IL-1, were significantly increased after renal I/R injury. In 2019, enhancer of zeste homolog 2 was found to be able to block nox4-dependent ROS production through the ALK5/SMad2/3 signaling pathway both in vitro and in mouse renal I/R injury models, thus inhibiting the canonical pyroptotic pathway of renal cells and attenuating renal I/R injury [95]. In the same year, Tajima et al. found that b-hydroxybutyrate can also mitigate renal I/R injury through its anti-pyroptotic effects [96]. In addition, Diao et al. suggested that inhibition of PRMT5 can activate the Nrf2/HO-1 signaling pathway and attenuate oxidative stress-induced pyroptosis in a mouse renal I/R injury model [97]. Similarly, it has been reported that by targeting the Nrf2 pathway, salvianolic acid B can also modulate caspase-1-mediated pyroptosis, thus alleviating renal I/R injury [98]. However, Xiao et al. indicated that the transcript induced in spermiogenesis 40 (Tisp40) had quite the opposite effect, inducing tubular epithelial cell GSDMD-mediated pyroptosis in renal I/R injury via NF-κB signaling [99]. By contrast, cholecalciferol pretreatment can protect renal function in I/R-induced GSDMD-mediated acute kidney injury through inhibiting NF-κB activation and reducing ROS production [100]. In addition, it seems that microRNA is also involved in renal I/R injury. Wang et al. identified microRNA-92a-3p as an essential regulator of tubular epithelial cell pyroptosis during renal I/R injury by targeting the Nrf1/HO-1 signaling pathway [101]. Furthermore, unlike aquaporin 4, which is involved in myocardial I/R injury, it is aquaporin 2 that is involved in renal I/R injury. Additionally, it has been illuminated that overexpression of aquaporin 2 can lighten the extent of pyroptosis of renal tubular epithelial cells during renal I/R injury [102]. While all of the aforementioned studies indicated that GSDMD acted as an executor and contributor to renal I/R injury, a recent study by Tonnus et al. found that GSDMD-deficient mice are hypersensitive to I/R-induced acute kidney injury (AKI) and demonstrated that GSDMD functioned as a suppressor of I/R-induced AKI through a previously unknown non-cell autonomous crosstalk to necroptosis [103]. Nonetheless, all these results shed light on the role of pyroptosis in renal I/R injury and may provide new insights into the treatment of renal I/R injury.

### 4.4. Pyroptosis and Hepatic I/R Injury

Liver I/R injury usually occurs during liver resection and liver transplantation, and it is the main cause of graft dysfunction and liver failure after liver transplantation. A study conducted by Li et al. suggested that DHA alleviates hepatic I/R injury by activating the PI3K/Akt pathway and subsequently inhibiting the canonical pyroptotic pathway [104]. Similarly, N-acetyl-1-tryptophan can lighten hepatic I/R injury via inhibiting the TLR4/NLRP3/caspase-1 signaling pathway [105]. Moreover, hepatic I/R injury was attenuated in caspase-1/caspase-11 double-gene knockout mice, suggesting that apart from the canonical pyroptotic pathway, the non-canonical pathway of pyroptosis is also involved in liver I/R injury [106]. Hua et al. reported that glycyrrhizin can suppress GSDMD-mediated pyroptosis of Kupffer cells to alleviate hepatic I/R injury [107]. Also targeting the pyroptosis of Kupffer cells, Wang et al. found that carbon monoxide-releasing molecule-3 could inhibit Kupffer cell pyroptosis via the sGC-cGMP signaling pathway to alleviate hepatic ischemia-reperfusion injury [108]. In addition, in steatotic livers, I/R injury induces pyroptosis and aggravates hepatic injury through caspase-1 activation [109]. Furthermore, it has been suggested that the stimulator of interferon genes (STING) could exacerbate liver I/R injury by facilitating calcium-dependent caspase-1-GSDMD processing in macrophages [110]. Finally, Alaa et al. found that the application of octreopeptide and melatonin can reduce inflammasome-induced pyroptosis by the iTLR4-NF-κB-NLRP3 pathway, thus attenuating hepatic I/R injury [111]. These results suggest that the canonical and non-canonical pathways of pyroptosis can be used as potential therapeutic targets for liver I/R injury.

### 4.5. Pyroptosis and Other I/R Injuries

Jia et al. established that metformin can protect against intestinal I/R injury and cell pyroptosis via the TXNIP-NLRP3-GSDMD pathway [112]. In the visual organ, the long noncoding RNA H19 is involved in initiating the pyroptosis of retinal microglia caused by retinal I/R injury, suggesting that pyroptosis also exists in retinal I/R injury [113]. In the respiratory organ, pretreatment with recombinant high-mobility group box 1 protein (rHMGB1) inhibited pyroptosis of alveolar macrophages through the Keap1/Nrf2/HO-1 signaling pathway, thus fighting against lung I/R injury [114]. In addition, it has been suggested that matrix metallopeptidase 2 (MMP2) and matrix metallopeptidase 9 (MMP9) contribute to lung I/R injury via the promotion of lung pyroptosis [115]. In summary, all the above results provide new insights into inhibiting pyroptosis in the treatment of I/R injury in these organs.

## 5. Conclusions and Perspective

In the past few decades, the treatments for various types of I/R injury have been far from satisfactory, and one of the main reasons lies in the complexity of the types of cell death, which include apoptosis, necrosis, autophagy, and the newly uncovered ferroptosis and pyroptosis [116]. Although it has been confirmed that pyroptosis is involved in the I/R injury of various organs, whether it is a major contributor to the I/R injury of any specific organ remains to be determined. Because different cell death types are not completely isolated and may have a synergistic effect on the progression of I/R injury [117], future studies exploring the interactions between different types of cell death and identifying medicines that could simultaneously regulate multiple targets are urgently required. In this context, for instance, Yu et al. reported that the Shexiang Baoxin Pill (SBP) could mitigate myocardial I/R injury by interfering with autophagy and inhibiting myocardial pyroptosis at the same time [118]. Given the fact that caspase-3 plays a pivotal role both in apoptosis and GSDME-mediated pyroptosis, it is reasonable to deduce that there must be some interaction between pyroptosis and apoptosis during I/R injury. However, research investigating the relationship between pyroptosis and apoptosis with respect to I/R injury has remained extremely limited to date. Nonetheless, some researchers have recently developed the concept of PANoptosis in the study of infectious and inflammatory diseases, where pyroptosis, apoptosis, and necroptosis act together in a multimeric protein complex called a PANoptosome [119,120]. This allows all the components of PANoptosis to be regulated simultaneously. Thus, PANoptosis provides a new way to study the regulation of cell death in which different types of cell death may be simultaneously regulated. Furthermore, recent studies have shown that PANoptosis-like cell death occurs during retinal and cerebral I/R injury [121,122], which could lay a solid foundation for the discovery of effective medicines that could simultaneously inhibit pyroptosis, apoptosis, and necroptosis and substantially alleviate I/R injury. Here, it is worth noting that the protective function of the medicines described in this review are mostly limited to animals, and how these findings translate into clinical treatment is yet to be determined. In addition, the mechanisms of pyroptosis still need further investigation at present; other new pathways and factors regulating pyroptosis might also be unearthed in the future. In addition, yet-to-be discovered genes and pathways may yield novel treatment methods for I/R injury. It should also be noted that the discovery of key genes and pathways involved in I/R injury is far from enough and does not amount to better outcomes for the treatment of I/R injury, as these newly found genes and pathways might have a dual roles during I/R injury. For example, Zhang et al. discovered that although the knockout of GSDMD conferred resistance to the heart against reperfusion injury in the acute phase of I/R, the knockout of GSDMD exacerbated reperfusion injury in the chronic phase of I/R via the activation of PARylation and the consumption of NAD+ and ATP, thus resulting in cardiomyocyte apoptosis. This study thus indicated a multidirectional role of GSDMD in I/R injury and a correlation between apoptosis and pyroptosis [123].

In sum, pyroptosis, a newly uncovered type of regulated cell death, plays an important role in the pathogenesis of various I/R injuries and organ dysfunction. Suppressing pyroptosis could be an effective therapeutic method for I/R injury. However, understanding the mechanisms of pyroptosis still requires in-depth investigation. In the future, after the mechanisms underpinning pyroptosis and its interactions with other types of cell death are fully understood, combined therapy simultaneously regulating multiple pathways might be the most effective strategy for limiting the severity of I/R injury.

## Figures and Tables

**Figure 1 biomolecules-12-01625-f001:**
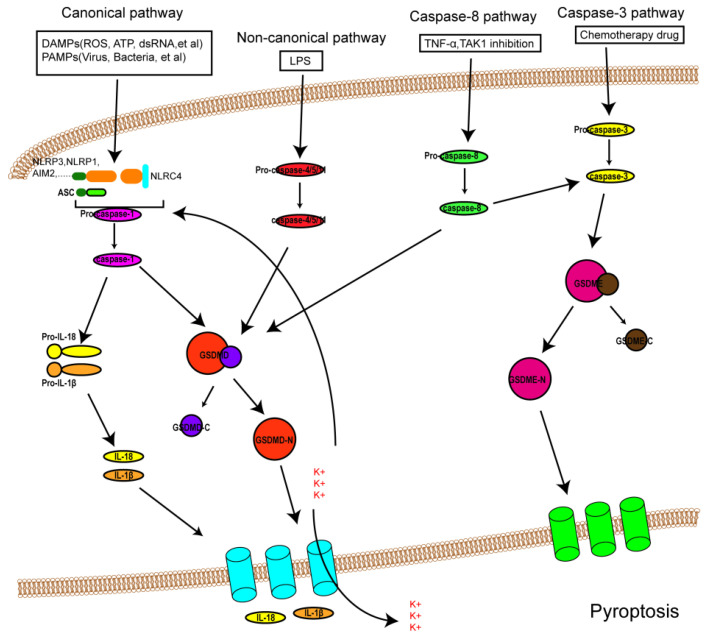
An overview of the mechanisms of pyroptosis. In the canonical pathway, PAMPs and DAMPs receive signaling molecule stimulation and assemble with pro-caspase-1 and ASC to form inflammasomes and active caspase-1. Activated caspase-1 cleaves GSDMD into GSDMD-N and GDSMD-C and lyses the precursors of IL-1β and IL-18 (pro-ILβ and pro-IL-18, respectively). GSDMD-N perforates the cell membrane by forming non-selective pores, further causing water influx and cell swelling and bursting. In addition, IL-1β and IL-18 are secreted from the pores formed by GSDMD-N. In the noncanonical pathway, LPS activates caspase-4/5 and caspase-11, triggering pyroptosis by cleaving GSDMD. In addition, the cleavage of GSDMD results in efflux of K+, ultimately giving rise to the assembly of an NLRP3 inflammasome. In the caspase-3-mediated pathway, active caspase-3 cleaves GSDME to form GSDME-N, inducing cell pyroptosis. In the caspase-8-mediated pathway, inhibiting TAK1 induces the activation of caspase-8, which cleaves GSDMD or GSDME and results in pyroptosis.

**Figure 2 biomolecules-12-01625-f002:**
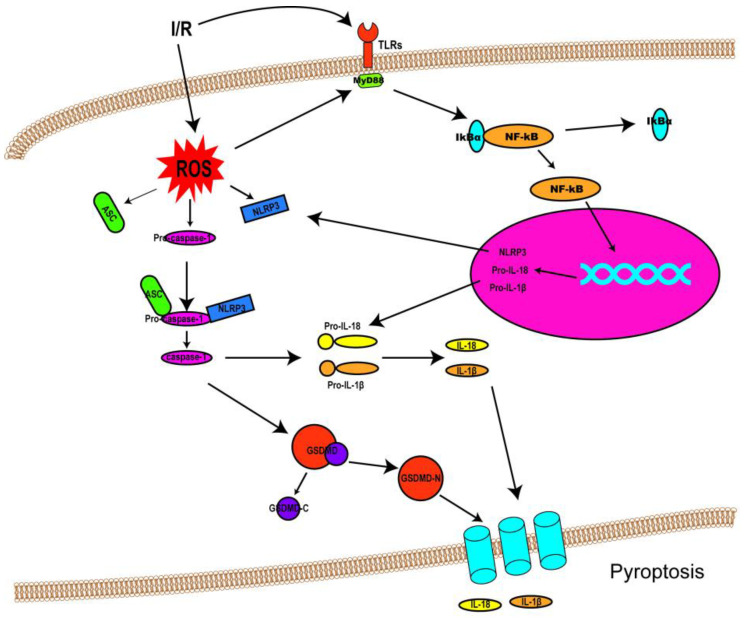
Schematic diagram demonstrating the link between pyroptosis and I/R injury. Following I/R insults, NF-κB signaling (priming) is activated by Toll-like receptors (TLRs), and ROS are also produced, resulting in NLRP3 inflammasome assembly and caspase-1 activation. Activated caspase-1 cleaves GSDMD into GSDMD-N and GDSMD-C and lyses the precursors of IL-1β and IL-18 (pro-ILβ and pro-IL-18, respectively) to form mature IL-1β and IL-18. Then, GSDMD-N oligomerizes and penetrates the cell membrane, forming non-selective pores on the membrane to release IL-1β and IL-18.

## Data Availability

Not applicable.

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
