# Peer review of "Pyroptosis: A Newly Discovered Therapeutic Target for Ischemia-Reperfusion Injury"

_biomolecules, 2022, doi:10.3390/biom12111625_

Round 1

Reviewer 1 Report

Yu Zheng et al summarizes the current status of pyroptosis and its connection with Ischemia-reperfusion of different organs, as well as the potential treatment strategies targeting it for Ischemia-reperfusion injury. Manuscript is well written and the question posed by the authors is well defined. The manuscript adheres to the relevant standards for reporting.  However, few suggestions to strengthen the manuscript.

In section 3, mechanisms of pyroptosis, one of the important mechanism, i.e. granzyme-mediated pyroptosis is not discussed in the manuscript.

The development of new molecules and drugs for Ischemia-reperfusion injury and treatment with respect to pyroptosis can be explained in detail.

The relationship between pyroptosis and apoptosis with respect to Ischemia-reperfusion injury can be discussed.

Reviewer 2 Report

The review by Zheng and colleagues is a nice overview of the latest findings of pytoptosis in IR injury and I found it enjoyable.

I have a few comments that may improve the manuscript.

Pyroptosis is not a newly discovered type of cell death. As the authors very well explain it was first identified in 1992. I would suggest that the authors rephrase the abstract

Gasdermin D-mediated pores and not non-selective. Indeed, they seem to be highly selective (Shiyu et al Nature 2021). I agree that this is not yet very clear but given such controversy it would be highly informative to discuss this.

The authors would possibly like to discuss the recent paper form Tonnus et al. Kidney int 2022 in section 4.3 about Gasdermins in kidney injury.

There are some spelling and grammar mistakes that the authors should proof-read (eg. Pyrophosis)
